# Evaluation of continuous arrhythmia monitoring using an implantable loop recorder in heart failure patients with a reduced ejection fraction: The LINQ2-HF trial rationale and protocol

Ryo Tateishi[1], Shinsuke Miyazaki[1*], Kazuya Yamao[1], Miho Negishi[1], Masaki Honda[1], Mari Omori[1], Yoshinori Kanno[1], Iwanari Kawamura[1], Yuji Matsuda[1], Yosuke Yamakami[1], Kentaro Goto[1], Kensuke Hirasawa[1], Takuro Nishimura[1], Mie Ochida[1], Tomoyo Sugiyama[1], Susumu Tao[1], Tomoyuki Umemoto[1], Masateru Takigawa[1], Taishi Yonetsu[1], Yasuhiro Maejima[1], Akihiro Hirakawa[2], Tetsuo Sasano[1]

**1** Department of Cardiovascular Medicine, Institute of Science Tokyo, Tokyo, Japan, **2** Department of Clinical Biostatistics, Institute of Science Tokyo, Tokyo, Japan

\* mmshinsuke@gmail.com

## Abstract

### Introduction

Heart failure with a reduced ejection fraction (HFrEF) is a common and serious condition often associated with atrial fibrillation (AF) and ventricular arrhythmias, leading to poor outcomes such as rehospitalizations and death. Detecting asymptomatic arrhythmias remains challenging, as traditional monitoring methods like 12-lead electrocardiograms (ECGs) or Holter ECGs are insufficient for continuous surveillance. This study aims to evaluate the utility of continuous arrhythmia monitoring using an implantable loop recorder (ILR) to detect asymptomatic arrhythmias in HFrEF patients, particularly for AF and ventricular tachycardia (VT).

### Methods and analysis

This is a single-center, non-randomized exploratory study. Thirty-five patients with HFrEF (LVEF ≤ 40%) and no history of AF will receive ILR implants. Primary endpoints include the detection of new-onset AF lasting at least 6 minutes and sustained or non-sustained VT. Secondary endpoints include the identification of bradycardia, pauses, and other arrhythmias, along with cardiovascular outcomes such as death and hospitalization. Additionally, secondary endpoints will include arrhythmia treatments, including anticoagulation therapy, catheter ablation, and implant of therapy devices. Data will be collected via remote monitoring through the Medtronic CareLink system, with event rates estimated using Kaplan-Meier methods. Data collection will span three years, with analyses conducted in the first and third years.

**Data availability statement:** As this is a study design paper, no datasets were generated or analyzed during the current study.

**Funding:** Institute of Science Tokyo is the sponsor of the study with financial support from Medtronic Japan Co., Ltd (ERP-2023-13609). The funders had no role in study design, data collection and analysis, decision to publish, or preparation of the manuscript.

**Competing interests:** Drs. Miyazaki, Takigawa, and Goto are affiliated to a department that received an unrestricted research grant from Medtronic.

**Abbreviations:** AF, Atrial fibrillation; CIED, Cardiac implantable electronic devices; ECG, Electrocardiogram; HF, Heart failure; HFrEF, Heart failure with reduced ejection fraction; ICD, Implantable cardioverter defibrillator; ILR, Implantable loop recorder; LVEF, Left ventricular ejection fraction; NYHA, New York Heart Association; VT, Ventricular tachycardia.

## Conclusion

The LINQ2-HF trial is designed to detect new-onset AF and VT in HFrEF patients using ILR, with the aim of informing future strategies for arrhythmia management in this population.

## Name of the registry

Japan Registry of Clinical Trials (jRCT).

## Trial registry number

jRCTs032240593

## Registration data

https://jrct.niph.go.jp/en-latest-detail/jRCTs032240593

---

## Introduction

Heart failure (HF) is a significant and growing public health concern globally, affecting millions of people worldwide. In Japan, the number of HF patients reached approximately 1.2 million in 2020, with expectations that this number will continue to rise due to the aging population [1]. Despite advancements in medical therapies, cardiac rehabilitation, and the development of cardiac implantable electronic devices (CIEDs), the prognosis for HF patients remains poor [2,3]. HF is associated with high rates of hospitalization, mortality, and significant healthcare costs.

Among heart failure patients, those with a reduced ejection fraction (HFrEF) account for nearly half of all HF cases. HFrEF patients often experience comorbid atrial fibrillation (AF) and ventricular arrhythmias. The coexistence of AF and HF has been shown to lead to worse outcomes compared to HF alone, largely due to the increased risk of thromboembolic events, hemodynamic compromise, and progression of HF [4]. Studies have indicated that early detection and intervention for AF, particularly in HF patients, can lead to improved clinical outcomes, including a reduced risk of a stroke and mortality [5,6]. Ventricular arrhythmias can lead to sudden cardiac death. Currently, the left ventricular ejection fraction (LVEF) is one of the strongest predictors of sudden cardiac death [7], with HFrEF patients at particularly high risk. Typically, a low ejection fraction combined with HF symptoms (New York Heart Association (NYHA) functional class II-III) is a key criterion for an implantable cardioverter defibrillator (ICD) implantation as primary prevention. However, some cases present with a low ejection fraction but no noticeable HF symptoms, making the NYHA classification assessment particularly challenging. This is especially problematic in the absence of documented ventricular arrhythmias, further complicating the decision to implant an ICD. While non-sustained ventricular arrhythmias also may sometimes be asymptomatic, non-sustained ventricular tachycardia (VT) is often an indication for an ICD implantation [8]. The

asymptomatic status of the patient not only complicates the detection of arrhythmias but also increases the likelihood that the patient may refuse an ICD implantation.

Traditional methods for arrhythmia detection, such as 12-lead electrocardiograms (ECGs) and Holter ECGs, are often limited by their brief monitoring windows. These methods frequently miss paroxysmal or asymptomatic AF and non-sustained VT episodes, leading to delayed diagnoses and suboptimal management of HFrEF patients. The introduction of implantable loop recorders (ILRs) offers a more comprehensive approach by providing continuous long-term heart rhythm monitoring [9]. ILRs have demonstrated higher efficacy in detecting intermittent arrhythmias compared to conventional monitoring techniques, particularly in patients at high risk for strokes and other cardiovascular events [10]. There is a previous study that demonstrates an ILR sensitivity of 95.2% and a specificity of 99.9%. It has been shown that ILR enables up to 14% more accurate assessment compared to ECG during symptomatic episodes or 24-hour ECG monitoring [11]. Their ability to capture asymptomatic arrhythmic episodes, such as silent AF or VT, allows for timely therapeutic interventions that could improve clinical outcomes.

The LINQ2-HF trial aims to evaluate the utility of continuous arrhythmia monitoring using an ILR, specifically focusing on detecting asymptomatic AF and VT in HFrEF patients. By identifying arrhythmias earlier in the disease course, ILR monitoring may enable more prompt and effective interventions, such as anticoagulation therapy for stroke prevention or rhythm control strategies to mitigate HF exacerbations [12]. Although some participants may be eligible for ICD implantation, certain individuals may be hesitant to proceed. In such cases, ILR findings may serve as objective evidence to support shared decision-making and promote more active engagement in treatment. Furthermore, early detection of AF may enable timely initiation of pharmacological therapy or catheter ablation before the onset of heart failure exacerbation or stroke. This study will also explore the potential for ILR-based monitoring to reduce rehospitalization rates and improve the overall survival in this high-risk patient population.

## Methods

### Study design

This is a prospective, single-center, non-randomized, and non-blinded exploratory study. The aim is to investigate the efficacy of continuous arrhythmia monitoring using an ILR in patients with HFrEF who have not been diagnosed with AF. The study will be conducted at a single medical facility, and all eligible patients will receive the same intervention. This study protocol was approved by the Institute of Science Tokyo Certified Review Board (NR2024−003). This study was conducted according to the guidelines of the Declaration of Helsinki.

### Participants

Participants in this study will be adults diagnosed with HFrEF, defined by a LVEF of 40% or less. Patients will be eligible for inclusion if they are receiving appropriate pharmacotherapy for HFrEF and have a $CHADS_2$ score of at least 1. Additionally, participants must be 20 years of age or older, and written informed consent must be obtained from each patient prior to enrollment in the study. The enrollment period is scheduled from January 2025 to January 2027.

Patients will be excluded from the study if they have already been diagnosed with AF or if they have previously been implanted with CIEDs such as pacemakers, ICDs, or ILRs. Other exclusion criteria include a life expectancy of less than one year, active infections, immunodeficiency, or the presence of other factors that, in the opinion of the principal investigator, may interfere with participation in the study or pose an undue risk to the patient. We have summarized the inclusion and exclusion criteria in Table 1.

Candidates will undergo a 12-lead ECG, echocardiogram, blood tests (including brain natriuretic peptide), and chest X-ray within one month prior to registration. After physicians verify these examinations and ensure that the inclusion and exclusion criteria are met, we will proceed with registration and the ILR implantations.

**Table 1. Inclusion and exclusion criteria.**

| Inclusion Criteria | Exclusion Criteria |
|---|---|
| 1) Patients with HFrEF (left ventricular ejection fraction ≤ 40%) | 1) Patients implanted with a CIED capable of detecting supraventricular arrhythmias (e.g., pacemaker, ICD, ILR) |
| 2) Patients receiving appropriate pharmacotherapy for HFrEF who can attend outpatient follow-up visits in the cardiology department at our hospital | 2) Patients diagnosed with AF at the time of consent |
| 3) Patients with a $CHADS_2$ score of 1 or higher | 3) Patients with an expected life prognosis of less than 1 year |
| 4) Patients aged 20 years or older at the time of consent | 4) Patients with immunodeficiency |
| 5) Patients who have provided written consent for participation in this clinical study | 5) Patients with active infections |
|  | 6) Patients with thin subcutaneous tissue at the implantation site, making ILR implantation unsafe |
|  | 7) Inappropriate case judged by doctor in charge |

AF, atrial fibrillation; CIED, cardiac implantable electronic device: ICD, implantable cardioverter defibrillator; ILR, implantable loop recorder; HFrEF, heart failure with reduced ejection fraction.

## ILR implantation

Eligible participants will undergo an implantation of an ILR (Reveal LINQII™, Medtronic, Minneapolis, MN) under local anesthesia. After making an approximately 1 cm incision in the left anterior chest, the device will be inserted subcutaneously using a minimally invasive procedure. Once the device is inserted, we will check the patients' hemostasis and close the surgical incision by a suture. Considering the battery life, the ILRs will be removed after three years of implantation.

## Follow-up

After implantation, the ILR will continuously monitor the patient's heart rate, and data will be transmitted remotely to the clinical team via the Medtronic CareLink system. This remote monitoring will continue for up to three years following the procedure, with the aim of detecting any episodes of arrhythmias, including AF and VT. Data from the ILR remote monitoring system will be reviewed and validated by two board-certified cardiologists.

The patients will attend cardiologist outpatient visits every one to three months, with regular remote monitoring checks every six months after the ILR implantation for three years. At each visit, data from the ILR will be reviewed, and additional tests, including ECGs and Holter ECGs, will be conducted to evaluate the accuracy and comprehensiveness of the ILR in detecting arrhythmic events. The schedule of enrollment, interventions, and assessments is shown in Fig 1. Lifestyle factors, including diet and smoking history, will also be reviewed to the extent that information is available in the electronic medical record.

## Endpoints

The primary endpoint of this study will be the detection of new-onset AF lasting at least six minutes or non-sustained or sustained VT. AF is defined as any episode lasting six minutes or longer. VT will be classified based on the duration and heart rate: non-sustained VT is defined as an episode lasting between 16 beats and 30 seconds with a heart rate exceeding 150 beats per minute, while sustained VT is defined as an episode lasting more than 30 seconds.

The secondary endpoints include: (1) the occurrence of bradycardia, defined as a heart rate below 30 beats per minute or pauses lasting more than 4.5 seconds, (2) the occurrence of other arrhythmias, such as high-degree atrioventricular block, and clinical events requiring a h4rpacemaker implantation or antiarrhythmic drug intervention, (3) cardiovascular

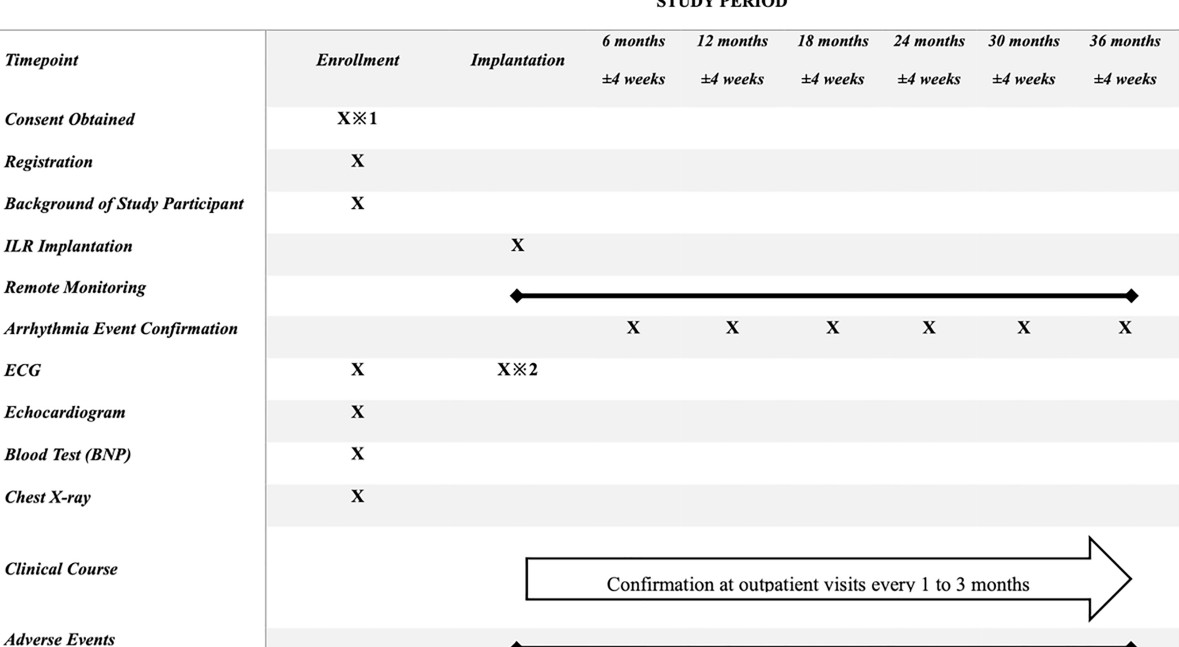

STUDY PERIOD

| Timepoint | Enrollment | Implantation | 6 months ±4 weeks | 12 months ±4 weeks | 18 months ±4 weeks | 24 months ±4 weeks | 30 months ±4 weeks | 36 months ±4 weeks |
|---|---|---|---|---|---|---|---|---|
| Consent Obtained | X※1 | | | | | | | |
| Registration | X | | | | | | | |
| Background of Study Participant | X | | | | | | | |
| ILR Implantation | | X | | | | | | |
| Remote Monitoring | | | ◆———————————————————————◆ | | | | | |
| Arrhythmia Event Confirmation | | | X | X | X | X | X | X |
| ECG | X | X※2 | | | | | | |
| Echocardiogram | X | | | | | | | |
| Blood Test (BNP) | X | | | | | | | |
| Chest X-ray | X | | | | | | | |
| Clinical Course | | | Confirmation at outpatient visits every 1 to 3 months ———▶ | | | | | |
| Adverse Events | | | ◆———————————————————————◆ | | | | | |

**Fig 1. The overview of the schedule for the LINQ2-HF trial. Additional Notes:** ※**1:** All investigations and observations related to this clinical study will be conducted after obtaining written informed consent and case registration from the study participants. Results from investigations, observations, and tests performed as part of routine clinical care prior to obtaining consent may be used as research data if conducted within 30 days prior to registration. ※**2:** As the baseline ECG, the rhythm will be confirmed using a monitoring ECG at the time of ILR implantation. BNP, brain natriuretic peptide; ECG, electrocardiogram; ILR, implantable loop recorder.

events, including all-cause mortality, cardiovascular death, hospitalization due to HF exacerbation, and strokes, (4) adverse events related to the ILR implantation, and (5) the rate of ILR malfunctions. The follow-up process and endpoint confirmation are summarized in Fig 2.

For the safety analysis, adverse events will be collected. An adverse event is defined as any unfavorable or unintended sign, symptom or disease.

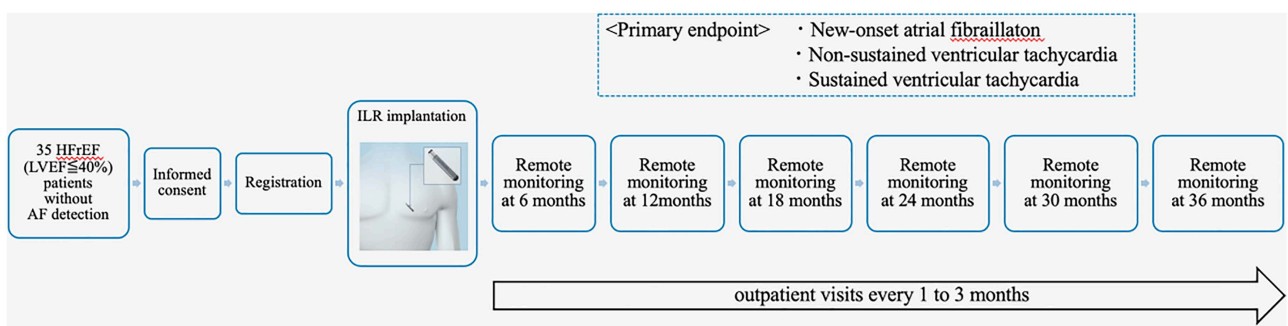

**Fig 2. Study flowchart of the LINQ2-HF trial.** AF, atrial fibrillation; HFrEF, heart failure with reduced ejection fraction; ILR, implantable loop recorder.

## Statistical analysis

In Japan, the use of ILRs in this study constitutes off-label use and is not covered by national health insurance. Therefore, device provision by Medtronic was necessary. Taking into account both the number of devices Medtronic could provide and the number of patients that could realistically be enrolled during the two-year recruitment period, the sample size was set at 35 participants for this exploratory study.

The primary endpoint is whether the binary outcome of new-onset AF or VT is detected by the ILR. This endpoint will be analyzed using the Kaplan–Meier method to estimate the incidence over the one-year monitoring period.

Secondary outcomes, including the occurrence of bradycardia, pauses, and other arrhythmic events, will be summarized using descriptive statistics. Correlations between arrhythmia detection and clinical outcomes, such as mortality and hospitalizations, will also be explored as part of the secondary analyses.

## Discussion/Conclusion

HF remains a leading cause of morbidity and mortality despite advances in medical therapy and device-based interventions [13]. Arrhythmias, particularly AF and VT, are common in patients with HFrEF and contribute to worse clinical outcomes, including rehospitalizations, strokes, and sudden cardiac death [8,14]. However, the intermittent and asymptomatic nature of many arrhythmias presents a diagnostic challenge.

This study will explore the potential of ILRs to overcome these limitations. In cases where CIEDs have not been implanted, an ILR provides continuous, long-term heart rate monitoring with minimal invasiveness. By detecting asymptomatic arrhythmias early, ILRs may enable timely therapeutic interventions, such as the initiation of anticoagulation or rhythm control strategies, which could significantly improve patient outcomes [12,15].

If successful, this pilot study could pave the way for larger clinical trials aimed at demonstrating the efficacy of ILR monitoring in improving the prognosis and reducing adverse events in HFrEF patients. Especially in Japan, the indications for ILR are primarily limited to the evaluation of syncope and the investigation of embolic strokes of an undetermined source, with its use for detailed arrhythmia assessment remaining limited. Although this was a single-center study and its findings may not be immediately generalizable, the results could still help shape clinical guidelines for ILRs in HF patients, potentially extending their use beyond current indications.

In conclusion, continuous arrhythmia monitoring using an ILR represents a promising approach to improving the management and prognosis of patients with HFrEF. The results of this study will provide valuable insights into the potential role of ILRs in detecting asymptomatic arrhythmias and optimizing patient care.

## Supporting information

**S1 File. SPIRIT checklist.**
(DOCX)

**S2 File. Copy of the protocol that was approved by the ethics committee (Japanese).**
(DOCX)

**S3 File. Copy of the protocol that was approved by the ethics committee (English).**
(DOCX)

## Acknowledgments

We would like to thank the members of Health Science Research & Development Center (HeRD) at the Institute of Science Tokyo for their assistance in preparing this clinical research, and Mr. John Martin for his help in the preparation of the manuscript.

## Author contributions

**Conceptualization:** Shinsuke Miyazaki.

**Data curation:** Ryo Tateishi, Shinsuke Miyazaki, Kazuya Yamao.

**Investigation:** Miho Negishi, Masaki Honda, Mari Omori, Yoshinori Kanno, Iwanari Kawamura, Yuji Matsuda, Yosuke Yamakami, Kentaro Goto, Kensuke Hirasawa, Takuro Nishimura, Mie Ochida, Tomoyo Sugiyama, Susumu Tao, Tomoyuki Umemoto, Masateru Takigawa, Taishi Yonetsu, Yasuhiro Maejima.

**Methodology:** Akihiro Hirakawa.

**Writing – original draft:** Ryo Tateishi, Shinsuke Miyazaki.

**Writing – review & editing:** Ryo Tateishi, Shinsuke Miyazaki, Akihiro Hirakawa, Tetsuo Sasano.

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
