## [Decision Letter · Decision Letter 0]

19 Mar 2025

PONE-D-25-08074Evaluation of Continuous Arrhythmia Monitoring Using an Implantable Loop Recorder in Heart Failure Patients with a Reduced Ejection Fraction: The LINQ2-HF trial rationale and protocolPLOS ONE

Dear Dr. Miyazaki,

Thank you for submitting your manuscript to PLOS ONE. After careful consideration, we feel that it has merit but does not fully meet PLOS ONE’s publication criteria as it currently stands. Therefore, we invite you to submit a revised version of the manuscript that addresses the points raised during the review process.

We look forward to receiving your revised manuscript.

Kind regards,

Amirmohammad Khalaji

Academic Editor

PLOS ONE

“Institute of Science Tokyo is the sponsor of the study with financial support from Medtronic Japan Co., Ltd (ERP-2023-13609).”

“None declared.”

4. In the online submission form, you indicated that [The data are available from the corresponding author upon reasonable request.].

Reviewers' comments:

Reviewer's Responses to Questions

**Comments to the Author**

1. Does the manuscript provide a valid rationale for the proposed study, with clearly identified and justified research questions?

Reviewer #1: Yes

Reviewer #2: Yes

Reviewer #3: Yes

2. Is the protocol technically sound and planned in a manner that will lead to a meaningful outcome and allow testing the stated hypotheses?

Reviewer #1: Yes

Reviewer #2: Yes

Reviewer #3: Yes

3. Is the methodology feasible and described in sufficient detail to allow the work to be replicable?

Reviewer #1: Yes

Reviewer #2: Yes

Reviewer #3: Yes

4. Have the authors described where all data underlying the findings will be made available when the study is complete?

Reviewer #1: Yes

Reviewer #2: Yes

Reviewer #3: Yes

5. Is the manuscript presented in an intelligible fashion and written in standard English?

Reviewer #1: Yes

Reviewer #2: Yes

Reviewer #3: Yes

6. Review Comments to the Author

You may also provide optional suggestions and comments to authors that they might find helpful in planning their study.

Reviewer #1: the manuscript is a study protocol that explores the use of an implantable loop recorder (ILR) for continuous arrhythmia monitoring in patients with heart failure with reduced ejection fraction (HFrEF).

The study is described as an exploratory pilot study with a sample size of 35 participants. why your sample size is 35?

Have you considered comparing ILR monitoring with historical data or previously published studies to strengthen the analysis?

the study must discuss and include ILR Sensitivity and Specificity.

Reviewer #2: The LINQ2-HF trial protocol presents a well-structured and clinically relevant study investigating the utility of ILRs for continuous arrhythmia monitoring in HFrEF. Given the strong association between undetected AF and ventricular arrhythmias with poor cardiovascular outcomes, this study has significant potential to contribute to heart failure management.

However, several issues need to be addressed to improve the study

1. A comparison group using conventional ECG monitoring would provide stronger evidence of ILR effectiveness

2. Consider stratifying participants based on factors such as NYHA class, BNP/NT-proBNP levels, and prior heart failure hospitalizations to improve result interpretation.

3. It would also be valuable to assess and report lifestyle factors (e.g., diet, exercise, socioeconomic status) that may influence arrhythmia risk.

4. The protocol states that event rates will be estimated using Kaplan-Meier methods. While this is appropriate for time-to-event data, multivariable Cox regression analysis could further help adjust for potential confounders.

5. Will ILR findings be correlated with clinical outcomes such as hospitalization rates, stroke incidence, or mortality? Clarifying this would strengthen the study’s impact.

Reviewer #3: Interesting protocol.Some issues shoudl be added

1) in the introduction it should be added benefit of this device when compared with intracardiac monitoring (some of these patients may need ICD)

2) do you think that authors may exploit machine learning to improve detection of AF and also of monitoring of heart failire un these patients

3)sample size calcuilation is needed

4) it should be added who checked the loop recorder tracks

5) potential benefit should be more detailed

7. PLOS authors have the option to publish the peer review history of their article (what does this mean? ). If published, this will include your full peer review and any attached files.

**Do you want your identity to be public for this peer review?** For information about this choice, including consent withdrawal, please see our Privacy Policy .

Reviewer #1: No

Reviewer #2: No

Reviewer #3: **Yes: ** Fabrizio D'Ascenzo

---

## [Author Response · Author response to Decision Letter 1]

17 Apr 2025

Response to the editor and the reviewers’ comments

We thank the editor and the reviewers for the insightful comments. We tried our best to address the issues raised by the editor and the reviewers, which helped us improve the quality of our manuscript significantly. We hope that the manuscript is now suitable for publication in PLOS One.

1. Please ensure that your manuscript meets PLOS ONE's style requirements,

including those for file naming. The PLOS ONE style templates can be found at https://journals.plos.org/plosone/s/file?id=wjVg/PLOSOne_formatting_sample_main_body.pdf and https://journals.plos.org/plosone/s/file?id=ba62/PLOSOne_formatting_sample_title_authors_affiliations.pdf

Thank you for your comment. We have revised the manuscript to comply with the PLOS ONE style guidelines, including font size and bold formatting. As the number of corrections is extensive, we have not listed each one individually; however, all changes have been highlighted for clarity.

“Institute of Science Tokyo is the sponsor of the study with financial support from Medtronic Japan Co., Ltd (ERP-2023-13609).”

We thank your suggestion. We have added the following sentence to the Funding section as requested:

Page 14, Line 247-248: "The funders had no role in study design, data collection and analysis, decision to publish, or preparation of the manuscript."

“None declared.”

Thank you for your guidance. Based on your instruction, we have included the following statement in the Competing Interests section:

Page 14, Lines 230–231: " Drs. Miyazaki, Takigawa, and Goto are affiliated to a department that received an unrestricted research grant from Medtronic."

4. In the online submission form, you indicated that [The data are available from the

corresponding author upon reasonable request.].

We appreciate your feedback. However, we would like to clarify that this is a study protocol paper, and therefore no data have been collected or are available at this time.

Therefore, the Data Availability Statement has been revised as follows:

Page 14, Lines 254–255: “As this is a study design paper, no datasets were generated or analyzed during the current study.”

5. Your ethics statement should only appear in the Methods section of your

manuscript. If your ethics statement is written in any section besides the Methods, please move it to the Methods section and delete it from any other section. Please ensure that your ethics statement is included in your manuscript, as the ethics statement entered into the online submission form will not be published alongside your manuscript.

We appreciate your comment. We have moved the ethics statement to the Methods section in accordance with your instructions. The revised statement is now located as follows:

Page 7, Lines 115–117: "This study protocol was approved by the Institute of Science Tokyo Certified Review Board (NR2024-003). This study was conducted according to the guidelines of the Declaration of Helsinki."

In addition, the ethics statement was removed from the abstract and moved to the Methods section. The conclusion was revised accordingly.

Page 2, Lines 40–42: “Conclusion: The LINQ2-HF trial is designed to detect new-onset AF and VT in HFrEF patients using ILR, with the aim of informing future strategies for arrhythmia management in this population.” 

Reviewer #1:the manuscript is a study protocol that explores the use of an implantable loop recorder (ILR) for continuous arrhythmia monitoring in patients with heart failure with reduced ejection fraction (HFrEF).

1. The study is described as an exploratory pilot study with a sample size of 35 participants. why your sample size is 35?

We appreciate the reviewer’s important comment. As per your suggestion, we have revised the statistical analysis section to clarify the rationale for our sample size. The following text has been added:

Page 11, Lines 194–198: “In Japan, the use of ILRs in this study constitutes off-label use and is not covered by national health insurance. Therefore, device provision by Medtronic was necessary. Taking into account both the number of devices Medtronic could provide and the number of patients that could realistically be enrolled during the two-year recruitment period, the sample size was set at 35 participants for this exploratory study.”

2. Have you considered comparing ILR monitoring with historical data or previously published studies to strengthen the analysis?

Thank you for your suggestion. We have addressed this point by citing previous studies that demonstrated ILR monitoring provides superior accuracy compared to traditional methods such as 24-hour ECG.

Page 6, Lines 92–93: “It has been shown that ILR enables up to 14% more accurate assessment compared to ECG during symptomatic episodes or 24-hour ECG monitoring.”

3. The study must discuss and include ILR Sensitivity and Specificity.

We appreciate your helpful feedback. As suggested, we have added information regarding ILR sensitivity and specificity in the manuscript.

Page 6, Lines 91–92: “There is a previous study that demonstrates an ILR sensitivity of 95.2% and a specificity of 99.9%.”

Reviewer #2: The LINQ2-HF trial protocol presents a well-structured and clinically relevant study investigating the utility of ILRs for continuous arrhythmia monitoring in HFrEF. Given the strong association between undetected AF and ventricular arrhythmias with poor cardiovascular outcomes, this study has significant potential to contribute to heart failure management.

However, several issues need to be addressed to improve the study.

1. A comparison group using conventional ECG monitoring would provide stronger evidence of ILR effectiveness

Thank you for this valuable comment. Our study is exploratory, and the sample size is relatively small; therefore, no comparative analysis is planned. Based on the findings of this study, we intend to conduct a subsequent study with a larger sample size, incorporating a formal sample size calculation to enable comparative analysis.

2. Consider stratifying participants based on factors such as NYHA class, BNP/NT-proBNP levels, and prior heart failure hospitalizations to improve result interpretation.

We thank the reviewer for the suggestion. BNP levels are assessed during the initial screening, and CHADS₂ score ≥1 is part of the inclusion criteria. Since the CHADS₂ score reflects several heart failure-related parameters, relevant clinical stratification will be inherently incorporated into the study through these criteria.

3. It would also be valuable to assess and report lifestyle factors (e.g., diet, exercise, socioeconomic status) that may influence arrhythmia risk.

We appreciate this important suggestion. Although lifestyle factors are not mandatory in our data collection, patients are followed at cardiology outpatient clinics, and relevant lifestyle information will be reviewed and noted from medical records whenever available.

4. The protocol states that event rates will be estimated using Kaplan-Meier methods. While this is appropriate for time-to-event data, multivariable Cox regression analysis could further help adjust for potential confounders.

Thank you for your comment regarding statistical methodology. We have incorporated the following sentence into the manuscript to reflect this addition:

Page 11, Lines 200–201: “To further explore factors associated with arrhythmia development, multivariable Cox regression analysis will be performed to identify predictors associated with the occurrence of AF and VT.”

5. Will ILR findings be correlated with clinical outcomes such as hospitalization rates, stroke incidence, or mortality? Clarifying this would strengthen the study’s impact.

Thank you for pointing this out. We apologize for any lack of clarity in the original description. As mentioned in the section on secondary endpoints, cardiovascular events will be assessed, including all-cause mortality, cardiovascular death, hospitalization due to heart failure exacerbation, and strokes. These outcomes will be analyzed in relation to ILR findings to evaluate their clinical significance. 

Reviewer #3: Interesting protocol. Some issues should be added

1. in the introduction it should be added benefit of this device when compared with intracardiac monitoring (some of these patients may need ICD)

Thank you for this important perspective. As the reviewer rightly noted, some of the patients enrolled in this study may be candidates for ICD implantation. We have clarified this point by adding the following statement to the introduction.

Page 6, Line 100-103: “Although some participants may be eligible for ICD implantation, certain individuals may be hesitant to proceed. In such cases, ILR findings may serve as objective evidence to support shared decision-making and promote more active engagement in treatment.”

2. do you think that authors may exploit machine learning to improve detection of AF and also of monitoring of heart failure un these patients

Thank you for your valuable comment. We would like to note that LINQ II incorporates the world’s first AI platform for ILRs, which applies artificial intelligence algorithms to data transmitted to the CareLink™ Network. This technology significantly reduces false alerts for atrial fibrillation and pauses, without compromising sensitivity.

For further details, please refer to: https://europe.medtronic.com/xd-en/c/emea/cardiac-rhythm/linq-system.html#accurhythm

3. sample size calculation is needed

We appreciate this important comment. As also raised by other reviewers, we have addressed this issue by adding the following description to the manuscript.

Page 11, Lines 194–198: “In Japan, the use of ILRs in this study constitutes off-label use and is not covered by national health insurance. Therefore, device provision by Medtronic was necessary. Taking into account both the number of devices Medtronic could provide and the number of patients that could realistically be enrolled during the two-year recruitment period, the sample size was set at 35 participants for this exploratory study.”

4. it should be added who checked the loop recorder tracks

Thank you for this pertinent comment. We have revised the manuscript to include the following clarification.

Page 9, Lines 153–154:“Data from the ILR remote monitoring system will be reviewed and validated by two board-certified cardiologists.”

5. potential benefit should be more detailed

We thank the reviewer for this valuable comment. Since this point partially overlaps with comment 1, we have incorporated additional explanation into the same section of the introduction. In addition, considering the potential benefits of early detection of atrial fibrillation, we have added the following sentence to the manuscript.

Page 6, Lines 103-105: “Furthermore, early detection of AF may enable timely initiation of pharmacological therapy or catheter ablation before the onset of heart failure exacerbation or stroke.”

We appreciate in advance for your excellent comments on this paper. We hope that our new version could correctly answer all the questions and is significantly improved over the previous submission.

---

## [Decision Letter · Decision Letter 1]

22 May 2025

PONE-D-25-08074R1Evaluation of Continuous Arrhythmia Monitoring Using an Implantable Loop Recorder in Heart Failure Patients with a Reduced Ejection Fraction: The LINQ2-HF trial rationale and protocolPLOS ONE

Dear Dr. Miyazaki,

Thank you for submitting your manuscript to PLOS ONE. After careful consideration, we feel that it has merit but does not fully meet PLOS ONE’s publication criteria as it currently stands. Therefore, we invite you to submit a revised version of the manuscript that addresses the points raised during the review process.

We look forward to receiving your revised manuscript.

Kind regards,

Amirmohammad Khalaji

Academic Editor

PLOS ONE

Reviewers' comments:

Reviewer's Responses to Questions

**Comments to the Author**

1. Does the manuscript provide a valid rationale for the proposed study, with clearly identified and justified research questions?

Reviewer #2: Yes

Reviewer #3: Yes

2. Is the protocol technically sound and planned in a manner that will lead to a meaningful outcome and allow testing the stated hypotheses?

Reviewer #2: Yes

Reviewer #3: Yes

3. Is the methodology feasible and described in sufficient detail to allow the work to be replicable?

Reviewer #2: Yes

Reviewer #3: Yes

4. Have the authors described where all data underlying the findings will be made available when the study is complete?

Reviewer #2: Yes

Reviewer #3: Yes

5. Is the manuscript presented in an intelligible fashion and written in standard English?

Reviewer #2: Yes

Reviewer #3: Yes

6. Review Comments to the Author

You may also provide optional suggestions and comments to authors that they might find helpful in planning their study.

**Reviewer #2:**  This study presents a clearly structured and clinically relevant study protocol addressing a significant gap in the management of HFrEF, the underdiagnosis of asymptomatic arrhythmias. The rationale is compelling, and the use of implantable loop recorders (ILRs) is timely given the limitations of conventional monitoring. The authors also demonstrate responsiveness to reviewer feedback, strengthening the manuscript substantially in the revision process. There are several issues that need to be addressed:

• The authors explained why only 35 patients were included, but they still need to add a statistical power calculation or effect size estimate to strengthen the study's reliability.

• Without a comparison group (like standard ECG or Holter), it's hard to judge how effective the ILR really is. Using historical or matched data could help.

• Since this is a single-center study, the results may not apply widely. This limitation should be stated clearly.

• The study mentions AI use in arrhythmia detection but doesn't explain how its accuracy will be measured. Future studies should assess how well the AI performs.

• Lifestyle factors (like diet, exercise, smoking) were noted as important but aren't part of the formal data collection. Including them, even optionally, could provide useful insights.

• Define all abbreviations the first time they appear (e.g., NYHA is not explained in the text).

**Reviewer #3:**  All comments have been addressed and authors should be complimented for performing such an accurate rebuttal. I think it is worth of publication.

7. PLOS authors have the option to publish the peer review history of their article (what does this mean? ). If published, this will include your full peer review and any attached files.

**Do you want your identity to be public for this peer review?** For information about this choice, including consent withdrawal, please see our Privacy Policy .

Reviewer #2: No

Reviewer #3: **Yes: ** Fabrizio D'Ascenzo

---

## [Author Response · Author response to Decision Letter 2]

29 May 2025

Response to the editor and the reviewers’ comments

We thank the editor and the reviewers for the insightful comments. We tried our best to address the issues raised by the editor and the reviewers, which helped us improve the quality of our manuscript significantly. We hope that the manuscript is now suitable for publication in PLOS One.

Reviewer #2:This study presents a clearly structured and clinically relevant study protocol addressing a significant gap in the management of HFrEF, the underdiagnosis of asymptomatic arrhythmias. The rationale is compelling, and the use of implantable loop recorders (ILRs) is timely given the limitations of conventional monitoring. The authors also demonstrate responsiveness to reviewer feedback, strengthening the manuscript substantially in the revision process. There are several issues that need to be addressed:

• The authors explained why only 35 patients were included, but they still need to add a statistical power calculation or effect size estimate to strengthen the study's reliability.

Thank you for the suggestion. As previously noted, enrolling 35 consecutive patients represented the maximum feasible number at our center during the study period. Because our primary aim was to determine the real-world prevalence of AF and VT in patients with HFrEF—not to compare interventions or groups—we considered that a priori power or effect-size calculations—essential in comparative designs—were not applicable to this descriptive study. This is an exploratory study for future research.

• Without a comparison group (like standard ECG or Holter), it's hard to judge how effective the ILR really is. Using historical or matched data could help.

We appreciate the reviewer's important comment. We agree that an external benchmark is helpful. Reference 6 reports a 17 % prevalence of AF in HFrEF based on standard 12-lead ECG. We believe that this historical figure provides a useful point of comparison for the findings of our study.

• Since this is a single-center study, the results may not apply widely. This limitation should be stated clearly.

We appreciate your helpful feedback. As recommended, we have emphasized that the single-center nature of the study may restrict the generalizability of our findings as below.

Page 12, Lines 227–229:”Although this was a single-center study and its findings may not be immediately generalizable, the results could still help shape clinical guidelines for ILRs in HF patients, potentially extending their use beyond current indications.”

• The study mentions AI use in arrhythmia detection but doesn't explain how its accuracy will be measured. Future studies should assess how well the AI performs.

We thank the reviewer’s insight. The ILR used in this study is the Reveal LINQ™ system, which employs the same algorithm evaluated in Reference 11. Because that publication provides detailed validation data, we have assumed that its validated performance also applies to our study population.

• Lifestyle factors (like diet, exercise, smoking) were noted as important but aren't part of the formal data collection. Including them, even optionally, could provide useful insights.

We appreciate this suggestion. Where information is available in the electronic medical record, we will now collect relevant lifestyle factors where available.

Page 10, Lines 162–163: “Lifestyle factors, including diet and smoking history, will also be reviewed to the extent that information is available in the electronic medical record.”

• Define all abbreviations the first time they appear (e.g., NYHA is not explained in the text).

We are very sorry for the lack of description of abbreviation. All abbreviations have now been defined at their first appearance in the manuscript.

Page 5, Lines 74–76: “Typically, a low ejection fraction combined with HF symptoms (New York Heart Association (NYHA) functional class Ⅱ-Ⅲ) is a key criterion for an implantable cardioverter defibrillator (ICD) implantation as primary prevention.”

We appreciate in advance for your excellent comments on this paper. We hope that our new version could correctly answer all the questions and is significantly improved over the previous submission.

Reviewer #3: All comments have been addressed and authors should be complimented for performing such an accurate rebuttal. I think it is worth of publication.

We sincerely thank the reviewer for the encouraging feedback and for acknowledging how carefully we addressed every comment. We greatly appreciate the recommendation for publication.

---

## [Decision Letter · Decision Letter 2]

16 Jul 2025

Evaluation of Continuous Arrhythmia Monitoring Using an Implantable Loop Recorder in Heart Failure Patients with a Reduced Ejection Fraction: The LINQ2-HF trial rationale and protocol

PONE-D-25-08074R2

Dear Dr. Miyazaki,

We’re pleased to inform you that your manuscript has been judged scientifically suitable for publication and will be formally accepted for publication once it meets all outstanding technical requirements.

Kind regards,

Amirmohammad Khalaji

Academic Editor

PLOS ONE

Additional Editor Comments (optional):

Reviewers' comments:

Reviewer's Responses to Questions

**Comments to the Author**

1. Does the manuscript provide a valid rationale for the proposed study, with clearly identified and justified research questions?

Reviewer #2: Yes

Reviewer #3: Yes

2. Is the protocol technically sound and planned in a manner that will lead to a meaningful outcome and allow testing the stated hypotheses?

Reviewer #2: Yes

Reviewer #3: Yes

3. Is the methodology feasible and described in sufficient detail to allow the work to be replicable?

Reviewer #2: Yes

Reviewer #3: Yes

4. Have the authors described where all data underlying the findings will be made available when the study is complete?

Reviewer #2: Yes

Reviewer #3: Yes

5. Is the manuscript presented in an intelligible fashion and written in standard English?

Reviewer #2: Yes

Reviewer #3: Yes

6. Review Comments to the Author

You may also provide optional suggestions and comments to authors that they might find helpful in planning their study.

Reviewer #2: The manuscript has been revised accordingly, with all my comments carefully addressed and clarified.

Reviewer #3: all comments have been addressed and i have no further concern about this. Authors should be complimented for

7. PLOS authors have the option to publish the peer review history of their article (what does this mean? ). If published, this will include your full peer review and any attached files.

**Do you want your identity to be public for this peer review?** For information about this choice, including consent withdrawal, please see our Privacy Policy .

Reviewer #2: No

Reviewer #3: **Yes: ** Fabrizio D'Ascenzo

---

## [Editor Report · Acceptance letter]

PONE-D-25-08074R2

PLOS ONE

Dear Dr. Miyazaki,

I'm pleased to inform you that your manuscript has been deemed suitable for publication in PLOS ONE. Congratulations! Your manuscript is now being handed over to our production team.

Kind regards,

on behalf of

Dr. Amirmohammad Khalaji

Academic Editor

PLOS ONE